# Phytochemicals and Glioma: Results from Dietary Mixed Exposure

**DOI:** 10.3390/brainsci13060902

**Published:** 2023-06-02

**Authors:** Weichunbai Zhang, Ce Wang, Feng Chen, Yongqi He, Shuo Yin, Yue Peng, Wenbin Li

**Affiliations:** Department of Neuro-Oncology, Cancer Center, Beijing Tiantan Hospital, Capital Medical University, Beijing 100070, China; zwchunbai@163.com (W.Z.); 13146851017@163.com (C.W.); chenfeng@bjtth.org (F.C.); 122021010536@mail.ccmu.edu.cn (Y.H.); 18931843059@163.com (S.Y.); pengyue227@163.com (Y.P.)

**Keywords:** phytochemicals, glioma, weighted quantile sum regression, dose–response relationship, Bayesian kernel machine regression

## Abstract

The information about phytochemicals’ potential to prevent cancer is encouraging, including for glioma. However, most studies on phytochemicals and glioma mainly focused on preclinical studies. Their epidemiological studies were not sufficient, and the evidence on the dose–response relationship is usually limited. Therefore, this investigation examined the association between dietary phytochemical intake and glioma in Chinese adults. This case–control study was carried out in a hospital in China. Based on the dietary information obtained from the food frequency questionnaire, the researchers estimated the phytochemical intake of 506 patients with glioma and 506 controls. Compared with participants in the lowest tertile, the highest intakes of carotene, flavonoids, soy isoflavones, anthocyanin, and resveratrol were associated with a reduced risk of glioma. The WQS and BKMR models suggested that anthocyanin and carotene have a greater influence on glioma. The significant nonlinear dose–response associations between dietary phytochemicals and glioma were suggested using the restricted cubic spline function. According to this study on phytochemicals and glioma, higher intakes of carotene, flavonoids, soy isoflavones, anthocyanins, and resveratrol are linked to a lower risk of glioma. So, we might not be able to ignore how phytochemicals affect gliomas.

## 1. Introduction

Phytochemicals refer to multiple intermediate or secondary metabolites produced in plant energy metabolism. They are divided into many subcategories according to their chemical structure, such as polyphenols, alkaloids, and carotenoids [1,2]. As the name implies, these substances are a major factor in how plants look and smell, and are commonly present in fruits, vegetables, nuts, etc. [3]. In recent years, phytochemicals, as non-traditional nutrients, have been considered one of the main reasons for the protective effect of plant-based foods against chronic diseases [4]. Because phytochemicals are abundant in all kinds of foods and have few side effects, more and more phytochemicals play essential roles in diseases, such as cardiovascular disease [5], diabetes [6], cancers [7], and so on.

There has been interest in the connection between phytochemicals and cancer. They can control the cell cycle and proliferation of tumor cells as well as their occurrence by participating in various signal pathways [7]. Epidemiological research has demonstrated that phytochemicals significantly reduce the risk of developing many malignancies. In a case–control study involving 415 patients and 830 controls in the Korean population, Kim et al. discovered a negative correlation between total dietary carotenoids and the risk of gastric cancer in women (odds ratio (OR) = 0.56, 95% confidence interval (95% CI): 0.32–0.99) and higher dietary lycopene intake and the risk of gastric cancer (OR = 0.60, 95% CI: 0.42–0.85) [8]. Intake of dietary lycopene was related to a lower risk of prostate cancer, according to a meta-analysis of 42 studies (relative risk (RR) = 0.88, 95% CI: 0.78–0.98). Plasma exposure data also indicated that lycopene was negatively related to prostate cancer (RR = 0.88, 95% CI: 0.79–0.98) [9]. Feng et al. assessed the dietary flavonoid intake in the Chinese population through FFQs and found that the highest quartile of flavonoids could reduce the risk of breast cancer by 34% (OR = 0.66, 95% CI: 0.54–0.82) [10]. Although this association has not been found in digestive system cancers [11,12], individual flavonoid subclasses still showed a strong correlation with them. Wei et al. reported that moderate soy intake among Chinese women was not related to breast cancer (RR = 1.00, 95% CI: 0.81–1.22) based on the China Kadoorie Biobank study. However, their meta-analysis revealed that the risk of breast cancer decreased by 3% for every 10 mg/d increase in soy isoflavone intake [13]. Similar results were also discovered in resveratrol. Levi et al. evaluated grape intake to replace resveratrol intake and reported that grape intake was negatively associated with breast cancer in 369 Swiss women between 1993 and 2003 (OR = 0.39, 95% CI: 0.25–0.62), and resveratrol has been considered a promising cancer treatment [14].

Although these phytochemicals have shown different roles in all tumors, studies on phytochemicals and gliomas are rare. Gliomas are malignant brain tumors that mainly affect the nervous system. They are characterized by the distortion of glial cells, resulting in uncontrollable growth and development [15]. Currently, interventions for treating gliomas include surgical resection, radiotherapy, or chemotherapy. Although they temporarily relieve the disease, they still cannot stop its rapid spread and expansion in the brain [16,17]. Experimental evidence has shown that phytochemicals such as quercetin and resveratrol can inhibit cell activity and reduce the migration of glioma cells [18]. They can also reduce the migration, invasion, and rapid growth of gliomas by inhibiting the expansion of the cell cycle [19,20]. Although research on phytochemicals’ potential to prevent cancer is encouraging, epidemiological studies on gliomas are insufficient, and evidence regarding doses is typically scant [1]. Therefore, we quantified the association between five phytochemicals and glioma based on the dose–response relationship. Considering that dietary phytochemicals are not ingested as a single species [21], the association of dietary phytochemicals with glioma was assessed based on a co-exposure model to provide some evidence of phytochemicals preventing gliomas to quantitatively evaluate the effect of dietary phytochemicals on glioma in the Chinese population.

## 2. Materials and Methods

### 2.1. Study Population

Our methodology has been described in great detail in previous research [22]. In short, we carried out this case–control study at Beijing Tiantan Hospital during 2022. Participants for the case and control groups were chosen via convenient sampling by the inclusion criteria. Except for glioma, the study individuals had to be adults (older than 18) before being chosen, and no further inclusion criteria were used. According to the 2021 diagnostic criteria for central nervous system cancers, the cases were glioma patients who were simultaneously identified by neuro-oncologists and pathologists within a few months [23]. After reading the written materials and hearing the oral description of the study methodology, and with the participants’ permission, the pertinent information was gathered using a face-to-face questionnaire. Exclusion criteria mainly included major eating behavior changes (such as weight loss), hormone use, and other treatments that had an impact on nutrition, endocrine, digestive, and neurological diseases, abnormal energy expenditure (>5000 or 400 kcal/d), and pregnancy [22]. The control group was healthy residents from the community who were also screened according to the inclusion and exclusion criteria described above and matched 1:1 with the case population by age (±5 years) and sex. The Institutional Review Board of Beijing Tiantan Hospital, Capital Medical University authorized the study (No. KY2022-203-02).

### 2.2. Dietary Intake Assessment

The food frequency questionnaire (FFQ) was the primary tool to assess dietary consumption. The FFQ was utilized by researchers with medical backgrounds to gather data on the types and amounts of food consumed over the previous 12 months. The reproducibility and validity of this FFQ have been verified in previous studies [22,24], and we have made simple modifications (adding and deleting several foods) based on the actual needs of this study. A total of 114 questions were asked about their intakes of food items. Each food item’s consumption was evaluated in three ways: whether it was consumed, how frequently (daily, weekly, or monthly), and how much was consumed overall. To increase the accuracy of meal estimation, the researchers also gave the subjects photographs of various food amounts and qualities during the survey. For statistical analysis, this was transformed into the daily intake of all food types based on the consumption frequency and each intake filled out by the individuals.

### 2.3. Assessment of Phytochemical Intake

The intake of phytochemicals was mainly calculated according to the dietary intake of individuals and the China food composition table. According to the intake frequency and intake of each food reported by the study subjects, the consumption of all foods was converted into the daily intake (g or mL). The China Food Composition Table provided the unit amounts of carotene, flavonoid, soy isoflavone, anthocyanin, and resveratrol in each food [25]. The total intake of these phytochemicals was calculated by multiplying the daily intake of each food by the unit amount of each phytochemical in the food and then summing the intake of each phytochemical in the different foods.

Among them, flavonoids included quercetin, myricetin, luteolin, kaempferol, and apigenin; soy isoflavones included daidzein, glycitein, and genistein, anthocyanins included delphinidin, cyamidin, and peonidin; and resveratrol included resveratrol and polydatin.

### 2.4. Other Variables

A questionnaire survey was used to gather all necessary data. Age, household income, education level, occupation, disease history (allergies, cancer, head trauma), and lifestyle habits were all asked about in the questionnaire (including smoking status and physical activity). The International Physical Activity Questionnaire was used by the individuals to rate their recent physical activity. The physical activity was then assessed and computed as a metabolic equivalent [26]. Qualified professionals determined the body mass index (BMI) during the survey using calibrated tools to assess weight and height. Additionally, over the past ten years, living close to electromagnetic fields and broadcast antennas has been categorized as residing in high-risk residential zones, which was also thought to be a potential complicating factor [27].

### 2.5. Statistical Analysis

To compare differences between the two groups, *t*-tests and chi-square tests were performed after describing the research population’s basic features. The ORs and 95% CIs of gliomas were estimated using the logistic regression model. Confounding variables were not adjusted in the rough model (Model 1). The multivariate model (Model 2) was modified to take into account all demographic information, disease history, and dietary preferences. To account for potential confounding variables, age, BMI, and energy intake were included as continuous variables. Other variables were considered classified variables.

Based on the subgroups organized by confounding factors, the multivariate model’s logistic regression was used for the subgroup analysis. Sensitivity analysis was performed by calculating the phytochemical intake per unit body weight and repeated logistic regression to avoid the influence of body weight difference. Additionally, to get around the inherent limitations of using phytochemicals as grade variables, the restricted cubic spline function was used to explore the dose–response relationship in Model 2, in which four nodes were located in the 20th, 40th, 60th, and 80th percentiles of phytochemical intake, and the 10th percentile was used as the reference group (OR = 1) [28].

To further investigate the relationship between phytochemical combinations and glioma, we assessed it using weighted quantile sum (WQS) regression analysis [29,30]. According to the bootstrap sampling, each phytochemical was assigned a weight, and the sum of the weights was 1. The weight of each component in the mixture reflected the contribution of that component. Specifically, 40% and 60% of random samples were used in this study to test and verify the data, and the bootstrap setting was 1000. Additionally, it was explored from the two directions of positive correlation and negative correlation.

We also used Bayesian Kernel Machine Regression (BKMR) to assess the association between the combined effects of these phytochemicals and glioma. BKMR model analyzed the nonlinear and non-additive exposure–response relationship through iterative regression and Bayesian algorithm [31,32]. The model used a Markov Chain Monte Carlo algorithm with 10,000 iterations using a Gaussian kernel. By comparing the estimated effects of all phytochemicals for a particular percentile with the estimated effects of all phytochemicals for the 50th percentile, we were able to assess the total impact of the consumption of the five phytochemicals on glioma. While maintaining the consumption of other phytochemicals at the median, the exposure-response function was used to investigate the connection between specific phytochemicals and gliomas. Finally, the interaction between the effects of any two phytochemicals on glioma was investigated and the effect of different quantiles of one phytochemical on the association between the other phytochemical and glioma was evaluated by establishing a bivariate pairwise exposure–response function. The phytochemical’s likelihood of being included in the model was computed as a conditional posterior inclusion probability (condPIP) by the model.

R 4.1.1 and SPSS 26.0 were used for all statistical analyses. The statistical significance level for all bilateral statistical tests was *p* < 0.05.

## 3. Results

### 3.1. Characteristics of the Study Population and Phytochemicals

The study involved 506 participants in the case group out of a total of 1012 participants. The two groups had a comparable age distribution (case: 42.62 ±13.09 years, control: 41.15 ± 12.85 years), identical sex distribution, and no differences in high-risk residential areas or history of head trauma. However, patients with gliomas had higher BMIs, lower educational levels, greater levels of smoking and physical activity, fewer allergies, and higher rates of cancer in their families than non-glioma patients. Additionally, there were variations in family income and occupation. However, regarding BMI (*p*= 0.095), occupation (*p*= 0.354), and history of allergic diseases (*p*= 0.471) in the male population, there was no distinction between them. In the female population, there was no difference in smoking (*p*= 0.308) and family history of cancer (*p*= 0.078). Other conditions were consistent with the whole population (Table 1).

In terms of phytochemicals, as shown in Figure 1, the intake of phytochemicals was substantially higher in the controls than in the cases. Additionally, the analysis of phytochemicals revealed significant correlations between them in Appendix A (Spearman correlation coefficients range from 0.386 to 0.815).

### 3.2. Association between Phytochemicals and Glioma

Table 2 displays the relation between five phytochemicals and gliomas. After adjusting for other variables (Model 2), the highest carotene consumption was related to a 93% reduced risk of glioma (OR = 0.07, 95% CI: 0.03–0.14), the highest flavonoid consumption was related to an 89% reduced risk of glioma (OR = 0.11, 95% CI: 0.05–0.23), the highest soy isoflavone consumption was related to a 71% reduced risk of glioma (OR = 0.29, 95% CI: 0.16–0.52), the highest anthocyanin consumption was related to a 95% reduced risk of glioma (OR = 0.05, 95% CI: 0.02–0.10), and the highest resveratrol consumption was related to an 82% reduced risk of glioma (OR = 0.18, 95% CI: 0.10–0.35). For the continuous variables of phytochemicals, the results were similar to the above.

### 3.3. Phytochemicals and Pathological Classification and Grading of Glioma

The analysis of glioma’s pathological classifications revealed that phytochemicals affected various glioma subtypes in distinct ways. For astrocytoma, carotene (OR = 0.33, 95% CI: 0.16–0.67), flavonoids (OR = 0.77, 95% CI: 0.65–0.92), soy isoflavones (OR = 0.77, 95% CI: 0.66–0.91), anthocyanins (OR = 0.71, 95% CI: 0.57–0.88), and resveratrol (OR = 0.50, 95% CI: 0.30–0.85) were significantly related to a significantly reduced risk. For glioblastoma, carotene (OR = 0.72, 95% CI: 0.61–0.85), flavonoids (OR = 0.85, 95% CI: 0.77–0.94), soy isoflavones (OR = 0.88, 95% CI: 0.81–0.95), and anthocyanins (OR = 0.82, 95% CI: 0.74–0.91) were related to a significantly reduced risk. Due to the small sample size of oligodendrogliomas, no further analysis was conducted (Table 3).

The results of phytochemicals and different grades of glioma showed that carotene (OR = 0.23, 95% CI: 0.08–0.62), flavonoids (OR = 0.82, 95% CI: 0.70–0.95), and anthocyanins (OR = 0.69, 95% CI: 0.55–0.86) were related to a significantly reduced risk of low-grade gliomas. For high-grade gliomas, carotene (OR = 0.69, 95% CI: 0.60–0.80), flavonoids (OR = 0.80, 95% CI: 0.74–0.88), soy isoflavones (OR = 0.91, 95% CI: 0.86–0.96), anthocyanins (OR = 0.79, 95% CI: 0.72–0.86), and resveratrol (OR = 0.75, 95% CI: 0.63–0.90) were related to a reduced risk (Table 4).

### 3.4. Subgroup Analysis and Sensitivity Analysis

Model 2 was used to correct the remaining confounding factors after stratification by confounding factors. The findings demonstrated that the majority of the five phytochemical groupings corresponded to the general population. Due to the small sample size, individual subgroup results are only partially representative (Appendix A).

The results of sensitivity analysis showed that the significance of the results was still consistent with the original results after replacing the original intake with the phytochemical intake per unit body weight (Appendix A).

### 3.5. Dose–Response Relationship

We flexibly modelled and visually predicted the association between phytochemicals and the risk of glioma in Figure 2. When the daily consumption of carotenes exceeded 1186.16 μg, the risk tended to decline. Glioma risk remained largely unchanged once the intake exceeded 2363.0 μg/d (*p_-nonlinearity_
*= 0.0096). Similar to carotene, the incidence of glioma decreased with increased intake of flavonoids when the intake exceeded 52.48 mg/d, but was largely unchanged once the consumption exceeded 97.37 mg/d (*p_-nonlinearity_
*= 0.0433). For soy isoflavones, the risk increased at first and then decreased, and this downward trend gradually weakened when the intake exceeded 6.14 mg/d (*p_-nonlinearity_
*= 0.0109). For anthocyanins, when the intake exceeded 4.53 mg/d, the risk tended to decrease with increased intake. After the intake exceeded 8.73 mg/d, the risk was relatively stable (*p_-nonlinearity_
*< 0.0001). Resveratrol, similar to anthocyanins, also showed a significant L-shaped trend. When exceeding 572.39 μg/d of intake, the risk tended to decrease with increased intake. After exceeding 1085.51 μg/d of intake, the risk of glioma was largely stable (*p_-nonlinearity_
*< 0.0001).

### 3.6. Effects of Mixed Exposure of Phytochemicals on Glioma Based on WQS and BKMR

In the WQS analysis, we observed that the increase in mixed exposure to phytochemicals was associated with a decrease in glioma risk (OR = 0.22, 95% CI: 0.17–0.29). The weight chart shows that anthocyanins play a dominant role in mixed exposure (46.80%); that is, the relationship is mainly driven by anthocyanins, followed by carotene (26.30%) and soy isoflavones (12.40%). However, no positive correlation was found between the phytochemical mixture and glioma (Figure 3).

BKMR results showed that the condPIPs of five phytochemicals were greater than 0.5 (Appendix A). It suggested that they all participated in the joint exposure to glioma. Among them, the condPIP of anthocyanins, carotene, and soy isoflavones was 1.000, suggesting that they have the greatest contribution to the impact of glioma, which was consistent with the WQS results. The overall relationship between mixed exposure to phytochemicals and glioma is shown in the Figure 4. Compared with the 50th percentile, the risk of glioma showed a downward trend, and the overall impact was statistically significant. The changing trend of the exposure–response function of five phytochemicals is shown in Appendix A. Overall, when other phytochemicals were at were at their median intakes, carotene, soy isoflavones, and anthocyanins showed downward trends associated with glioma, while the trend of flavonoids and resveratrol tended to be stable. The bivariate paired exposure–response function showed that when carotene, soy isoflavone, and anthocyanin were fixed at the median intake, the slope of flavonoids changed with the increase in resveratrol from the 25th percentile to the 75th percentile, suggesting that there was an interaction between them (Appendix A).

## 4. Discussion

Our research examined the relationship between gliomas and five popular phytochemicals in the Chinese population. The findings demonstrated a substantial inverse relationship between carotene, flavonoid, soy isoflavone, anthocyanin, and resveratrol intake and glioma. Similar results were noted in various pathological subtypes and grades, particularly for the first time. Significant nonlinear dose–response correlations between these phytochemicals and glioma were further validated by the restricted cubic spline model. Glioma risk tended to stabilize after a certain intake was reached. The assessment of mixed exposure found that the mixture of phytochemicals was related to the reduction in glioma risk, in which anthocyanin and carotene contributed more, while flavonoids and resveratrol had some interaction.

Carotene is an important natural pigment that is yellow, red, or orange [33]. In recent years, carotene has been linked to some health advantages [5]. Because the body cannot synthesize carotene, it must be ingested through food or supplements [34]. Therefore, early epidemiological studies on carotene and cancer mainly relied on evaluating carotene-rich fruits and vegetables. They found that eating these plant foods can reduce the risk of some cancers, such as prostate cancer and lung cancer [35,36], but other nutrients in these foods may lead to confusion, such as dietary fiber and vitamins. The correlation between intake of carotene and cancer has been the subject of several investigations. These findings suggested that higher carotene intake had a protective effect against a variety of cancers, including prostate cancer (RR = 0.98, 95% CI: 0.96–0.99) [37], lung cancer (RR = 0.77, 95% CI: 0.68–0.87) [38], and esophageal cancer (OR = 0.62, 95% CI: 0.50–0.77) [39]. Compared with other phytochemicals, there were relatively more epidemiological studies on carotene and gliomas. Higher carotene intake can lower the incidence of glioma (OR = 0.65, 95% CI: 0.48–0.88), according to Tedeschi-Blok et al.’s findings [40]. Based on the Nebraska Health Study II, Chen et al. also discovered a related association. There was a strong correlation between two carotene subtypes—α-carotene (OR = 0.50, 95% CI: 0.30–0.80) and β-carotene (OR = 0.50, 95% CI: 0.30–0.90) and glioma [41], and carotene-rich foods were also negatively related to glioma (OR = 0.70, 95% CI: 0.50–0.90) [42]. These were consistent with our findings (OR = 0.07, 95% CI: 0.03–0.14), and we also found a nonlinear dose–response relationship in which the risk of glioma did not change when the intake of carotene exceeded 2363.0 μg/d. However, Holick et al. did not find this significant protective effect in three large American cohort studies, which was considered to be related to the different carotene subtypes assessed [43]. Moreover, it may be related to the metabolism of vitamin A. Some carotene can be transformed into vitamin A in the body [44]. According to a prior meta-analysis, vitamin A may lower the risk of glioma (RR = 0.80, 95% CI: 0.62–0.98) [45], and metabolites of vitamin A were also related to the prognosis of gliomas [46].

The most prevalent polyphenols found in food are flavonoids. They are ubiquitous in plant food, especially in the epidermis of fruits or the leaves of vegetables [47,48], with pronounced antiviral, antiallergic, anti-inflammatory, and antitumor activities [49,50]. The results of our investigation, which examined the relationship between flavonoid consumption and the risk of glioma, revealed a significant nonlinear dose–response relationship between increased flavonoid intake and glioma (OR = 0.11, 95% CI: 0.05–0.23). This was consistent with the results of a previous prospective study. Bever et al. found that the increased intake of flavan-3-oleopolymeric flavonoids in the diet was related to the reduced risk of glioma, especially flavonoids in tea [51].

Soy isoflavones are a natural compound from plants and have structures and functions similar to estrogen. The primary dietary source of isoflavones for human consumption is soybean and its products [52]. The mechanisms behind the antiobesity, hypoglycemic, and cancer-prevention properties of soy isoflavones have recently been explored [6]. Because of the antiestrogenic effect of soy isoflavones, most studies have focused on breast cancer [53]. Since there have not been many studies on soy isoflavones and gliomas, this epidemiological study based on the Chinese population may be the first to assess the connection between soy isoflavone intake and gliomas and has discovered that consuming more soy isoflavones can lower the risk of gliomas (OR = 0.29, 95% CI: 0.16–0.52). When the intake exceeded 30 mg/d, the risk of gliomas stabilized. DeLorenze et al. found patients with grade III gliomas who consumed the phytoestrogen secoisolariciresinol had a better cancer survival rate (HR = 0.48, 95% CI: 0.25–0.92) [54].

Anthocyanin is a naturally occurring pigment that is commonly present in plant leaves, flowers, and seeds [55], which is the main reason why plants show red, orange, purple, and blue [56]. Recent investigations on humans and animals discovered that anthocyanins can stop or delay the onset of some chronic diseases, including atherosclerosis [57], and can help with weight control [58]. Another study mainly focused on the digestive tract in the epidemiological study of cancer [59]. However, there have been few epidemiological studies on other cancers [60]. Our study found an L-shaped dose–response relationship between anthocyanin intake and glioma. This curve suggested that there was almost no change in the risk of glioma after the intake exceeded 8.73 mg/d (*p_-nonlinearity_
*< 0.0001). However, in another prospective study, no negative association was observed between them [51].

A type of polyphenol known as resveratrol is advantageous to human health and possesses several positive biological functions, including anti-inflammation, antioxidation, heart protection, and anticancer characteristics [61,62]. This compound is widely found in eucalyptus, pine, and lilies, but people’s intake is mainly from berries, nuts, and red grapes [63,64]. Although numerous studies reported that resveratrol can inhibit cancer cell proliferation and differentiation in the colon, prostate, breast, and lungs, most evidence was restricted to in vitro experiments and preclinical studies. Epidemiological studies are still lacking [65]. This was the first observational study to find the protective effect of resveratrol intake on glioma (OR = 0.18, 95% CI: 0.10–0.35). Additionally, its dose–response relationship was nonlinear, similar to the in vitro experiment on colorectal cancer [66]. The concentration of resveratrol in grape skin reaches 50–100 mg/g, which leads to more resveratrol in wine [67]. Studies on Mediterranean participants emphasize that moderate wine consumption also reduced the risk of glioma, which might also be related to resveratrol [68,69].

At present, the mechanism of phytochemicals in glioma is not clear. On the one hand, they participate in the regulation of apoptosis and autophagy. Coelho et al. also found that the flavonoid apigenin can decrease the survival rate and proliferation rate of rat C6 glioma cells in a time-dependent and dose-dependent manner by inducing differentiation, apoptosis, and autophagy [70]. Other phytochemicals, such as anthocyanins [71] and resveratrol [72], had similar mechanisms. On the other hand, they played an antioxidant protective role in scavenging DNA-damaged free radicals and regulating DNA repair mechanisms [73,74]. In rats given 50 mg/kg naringenin orally for 30 days, it was discovered that lipid peroxidation was decreased, antioxidant capacity was increased, and the expression of nuclear factor κB was decreased in rat brain tissue. These changes significantly slowed the proliferation of glioma cells [75]. Therefore, they may still have potential preventive and therapeutic effects on gliomas.

This study had the limitation of not being able to investigate the connection between the subtypes of five phytochemicals and gliomas. Previous in vitro studies have focused on certain phytochemical subtypes, such as quercetin, lignans, etc. However, we were unable to fully assess the impact of phytochemical subtypes on gliomas because the food composition table only supplied these five categories of phytochemicals. Secondly, our assessment of phytochemical intake only depended on the FFQ, which may be biased. However, we have minimized this bias through one-on-one interviews and food mapping assistance. Thirdly, there were significant differences in BMI, education, smoking, allergies, and family history of cancer between the case and control groups. These were considered to be related to other potential influencing factors of glioma. Existing studies have shown that glioma patients and healthy people were essentially different in smoking [68], BMI [76], allergies [77], family inheritance [78], and so on. This was similar to other study populations on diet and glioma [68]. To avoid potential confounding factors, we also adjusted these results in a multivariate model. Detailed subgroup analyses were performed for these variables, and the results remained relatively robust. In addition, we were unable to confirm a causal link between the two, and this case–control study’s intrinsic shortcomings cannot be avoided. However, there were still some benefits to this study. Firstly, this study investigated the connection between the consumption of five phytochemicals and glioma, and the findings were consistent with those of previous in vitro research, particularly for resveratrol and soybean isoflavones, which lacked clinical investigations. Moreover, it is the first time that the dose–response relationship between phytochemical consumption and glioma has been described, and the strong nonlinear dose–response association offers more epidemiological evidence for preventing glioma by phytochemicals. Importantly, this is the first time that the association with glioma has been assessed in the context of mixed phytochemical intake. As the main source of phytochemicals, people did not take a single phytochemical, but a variety of phytochemicals were taken together through the dietary route. Therefore, the effect of co-exposure to these phytochemicals on gliomas had to be considered. Based on WQS regression and the BKMR model, we found the association between phytochemical co-exposure and glioma and further found that anthocyanins and carotene had the greatest effect on glioma among many phytochemicals.

## 5. Conclusions

In conclusion, in this study of phytochemicals and glioma, we found that higher intakes of carotene, flavonoids, soy isoflavones, anthocyanins, and resveratrol were associated with a reduced risk of glioma, all of which had a significant dose–response relationship with glioma, and that the risk leveled off when intakes exceeded a certain amount. When exposure to a mixture of phytochemicals was considered, a significant association with glioma was still demonstrated and was primarily caused by anthocyanins and carotene. Therefore, the effect of phytochemicals on glioma may be too significant to ignore. Future prospective studies should further confirm this relationship.

## Figures and Tables

**Figure 1 brainsci-13-00902-f001:**
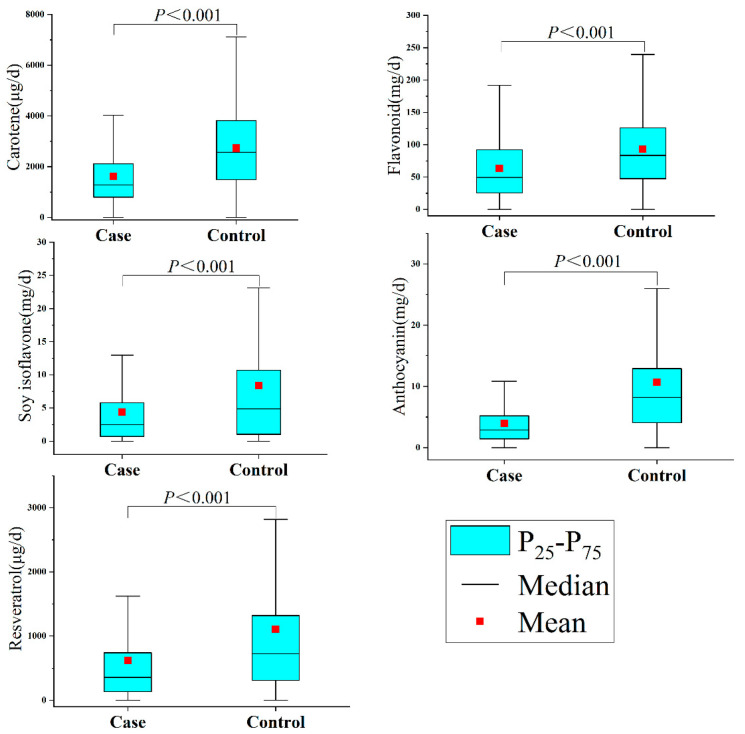
Phytochemical intakes among study participants.

**Figure 2 brainsci-13-00902-f002:**
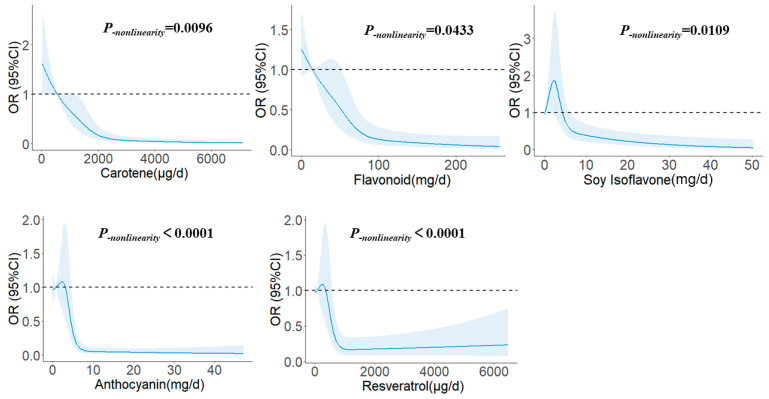
The restricted cubic spline for the associations between phytochemicals and glioma. The lines represent adjusted odds ratios based on restricted cubic splines for the phytochemical intakes in the regression model. Knots were placed at the 20th, 40th, 60th, and 80th percentiles of the phytochemical intakes, and the reference value was set at the 10th percentile. The adjusted factors were the same as in Model 2.

**Figure 3 brainsci-13-00902-f003:**
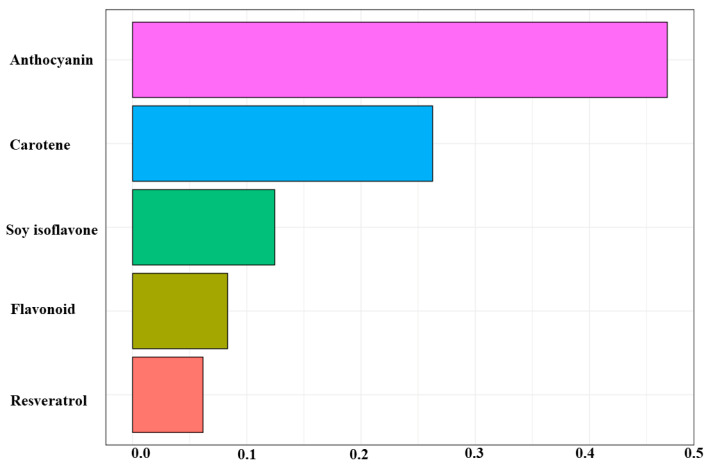
Results for the weighted quantile sum analyses.

**Figure 4 brainsci-13-00902-f004:**
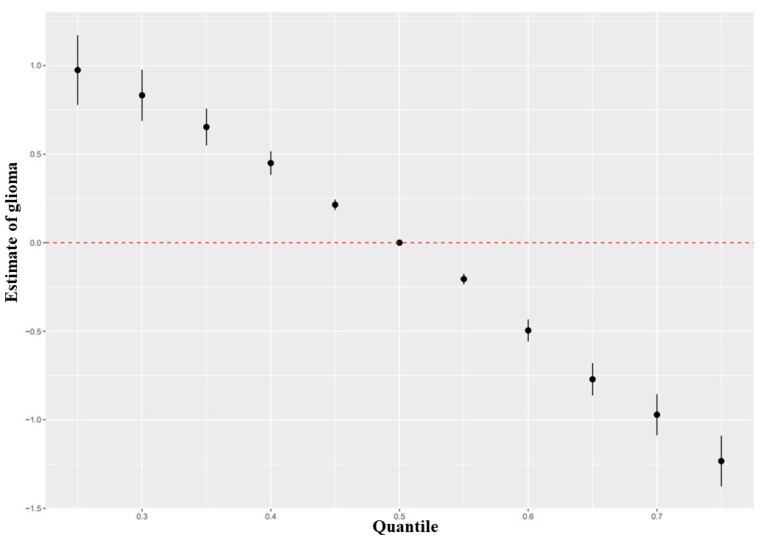
Effect of phytochemical mixture on glioma, comparing various percentiles of the mixture to the median (50th percentile).

**Table 1 brainsci-13-00902-t001:** Basic characteristics of the study participants.

	Male			Female			*p*-Value ^a,b^
	Case	Control	*p* Value ^a^	Case	Control	*p* Value ^a^	
Age (years)	42.29 ± 12.93	41.01 ± 12.83	0.235	43.04 ± 13.31	41.33 ± 12.90	0.171	0.072
BMI	24.60 ± 3.08	24.16 ± 3.21	0.095	23.30 ± 3.32	21.63 ± 2.76	<0.001	<0.001
High-risk residential area (%)			0.688			0.138	0.534
Yes	21.8	23.2		20.7	15.3		
No	78.2	76.8		79.3	84.7		
Occupation (%)			0.354			0.002	0.024
Manual workers	29.6	25.0		22.5	14.4		
Mental workers	59.5	61.3		43.3	59.5		
Others	10.9	13.7		34.2	26.1		
Education level (%)			0.003			<0.001	<0.001
Primary school and below	3.5	2.8		11.3	2.3		
Middle school	39.8	26.8		43.7	23.0		
University and above	56.7	70.4		45.0	74.7		
Household income (%)			<0.001			<0.001	<0.001
<3000 CNY/month	8.1	20.1		11.7	15.8		
3000–10,000 CNY/month	75.4	49.6		76.6	48.6		
>10,000 CNY/month	16.5	30.3		11.7	35.6		
Smoking status (%)			0.036			0.308	0.039
Never smoking	47.9	56.3		98.2	99.5		
Former smoking	22.2	14.4		0.9	0		
Current smoking	29.9	29.2		0.9	0.5		
History of allergies (%)			0.471			<0.001	<0.001
Yes	8.5	10.2		6.8	20.3		
No	91.5	89.8		93.2	79.7		
History of head trauma (%)			0.374			1.000	0.474
Yes	13.7	11.3		8.1	8.1		
No	86.3	88.7		91.9	91.9		
Family history of cancer (%)			0.005			0.078	0.001
Yes	27.1	17.3		33.8	26.1		
No	72.9	82.7		66.2	73.9		
Physical Activity (%)			<0.001			<0.001	<0.001
Low	16.2	45.1		10.4	46.8		
Moderate	44.7	34.8		36.9	38.3		
High	39.1	20.1		52.7	14.9		

^a.^ *p*-values were derived from Student’s *t*-tests for continuous variables according to the data distribution and the chi-square test for the classified variables. ^b.^ Results of the overall case group and the overall control group.

**Table 2 brainsci-13-00902-t002:** Adjusted ORs and 95% CIs for the association between phytochemicals and glioma.

Phytochemicals	T1	T2	T3	Continuous ^c^	*p*-Value
Carotene	≤1255.00	1255.00–2569.31	>2569.31		
Case/Control	247/91	174/171	85/244		
Model 1 ^a^	1	0.33 (0.22–0.48)	0.13 (0.08–0.19)	0.76 (0.72–0.81)	<0.001
Model 2 ^b^	1	0.29 (0.15–0.54)	0.07 (0.03–0.14)	0.67 (0.60–0.75)	<0.001
Flavonoid	≤42.82	42.82–93.63	>93.63		
Case/Control	223/115	158/179	125/212		
Model 1 ^a^	1	0.45 (0.33–0.62)	0.29 (0.20–0.40)	0.91 (0.88–0.93)	<0.001
Model 2 ^b^	1	0.40 (0.23–0.72)	0.11 (0.05–0.23)	0.85 (0.80–0.90)	<0.001
Soy isoflavone	≤1.66	1.66–6.33	>6.33		
Case/Control	189/149	208/129	109/228		
Model 1 ^a^	1	1.18 (0.85–1.64)	0.35 (0.24–0.49)	0.94 (0.92–0.96)	<0.001
Model 2 ^b^	1	1.10 (0.64–1.91)	0.29 (0.16–0.52)	0.92 (0.89–0.95)	<0.001
Anthocyanin	247/91	200/137	59/278		
Case/Control	≤2.76	2.76–7.57	>7.57		
Model 1 ^a^	1	0.54 (0.38–0.78)	0.07 (0.05–0.12)	0.84 (0.81–0.87)	<0.001
Model 2 ^b^	1	0.40 (0.21–0.74)	0.05 (0.02–0.10)	0.83 (0.78–0.87)	<0.001
Resveratrol	≤289.85	289.85–848.88	>848.88		
Case/Control	216/122	179/158	111/226		
Model 1 ^a^	1	0.60 (0.43–0.83)	0.27 (0.19–0.38)	0.79 (0.73–0.86)	<0.001
Model 2 ^b^	1	0.52 (0.29–0.90)	0.18 (0.10–0.35)	0.84 (0.75–0.95)	<0.001

^a.^ Model 1: unadjusted model; ^b.^ Model 2: adjusted for age, BMI, occupation, education level, household income, high-risk residential areas, smoking status, history of allergies, history of head trauma, family history of cancer, physical activity, and energy intake; ^c.^ carotene per 500 μg/d increments, flavonoid per 10 mg/d increments, soy isoflavone and anthocyanin per 1 mg/d increments, resveratrol per 500 μg/d increments.

**Table 3 brainsci-13-00902-t003:** Adjusted ORs and 95% CIs for the association between phytochemicals and gliomas of different pathological classifications.

Pathological Classification ^c^	Model 1 ^a^	*p* Value	Model 2 ^b^	*p*-Value
Astrocytoma				
Carotene	0.70 (0.60–0.81)	<0.001	0.33 (0.16–0.67)	0.003
Flavonoid	0.91 (0.86–0.96)	0.001	0.77 (0.65–0.92)	0.003
Soy isoflavone	0.90 (0.85–0.96)	0.001	0.77 (0.66–0.91)	0.002
Anthocyanin	0.74 (0.65–0.84)	<0.001	0.71 (0.57–0.88)	0.001
Resveratrol	0.67 (0.53–0.86)	0.001	0.50 (0.30–0.85)	0.011
Glioblastoma				
Carotene	0.79 (0.74–0.86)	<0.001	0.72 (0.61–0.85)	<0.001
Flavonoid	0.91 (0.88–0.95)	<0.001	0.85 (0.77–0.94)	0.001
Soy isoflavone	0.94 (0.91–0.97)	<0.001	0.88 (0.81–0.95)	0.001
Anthocyanin	0.86 (0.82–0.90)	<0.001	0.82 (0.74–0.91)	<0.001
Resveratrol	0.84 (0.75–0.93)	0.001	0.84 (0.69–1.04)	0.103

Note: No further analysis was conducted due to the small sample size of oligodendroglioma. ^a^. Model 1: unadjusted model. ^b^. Model 2: adjusted for age, BMI, occupation, education level, household income, high-risk residential areas, smoking status, history of allergies, history of head trauma, family history of cancer, physical activity, and energy intake. ^c^. Carotene per 500 μg/d increments, flavonoid per 10 mg/d increments, soy isoflavone and anthocyanin per 1 mg/d increments, resveratrol per 500 μg/d increments.

**Table 4 brainsci-13-00902-t004:** Adjusted ORs and 95% CIs for the association between phytochemicals and glioma of different grades.

Glioma Grading ^c^	Model 1 ^a^	*p* Value	Model 2 ^b^	*p*-Value
Low grade				
Carotene	0.71 (0.61–0.83)	<0.001	0.23 (0.08–0.62)	0.004
Flavonoid	0.91 (0.86–0.97)	0.002	0.82 (0.70–0.95)	0.009
Soy isoflavone	0.94 (0.90–0.99)	0.009	0.95 (0.89–1.02)	0.171
Anthocyanin	0.81 (0.74–0.89)	<0.001	0.69 (0.55–0.86)	0.001
Resveratrol	0.81 (0.69–0.95)	0.011	0.83 (0.62–1.11)	0.205
High grade				
Carotene	0.78 (0.73–0.83)	<0.001	0.69 (0.60–0.80)	<0.001
Flavonoid	0.91 (0.88–0.94)	<0.001	0.80 (0.74–0.88)	<0.001
Soy isoflavone	0.94 (0.91–0.97)	<0.001	0.91 (0.86–0.96)	0.001
Anthocyanin	0.84 (0.80–0.88)	<0.001	0.79 (0.72–0.86)	<0.001
Resveratrol	0.78 (0.70–0.87)	<0.001	0.75 (0.63–0.90)	0.002

^a.^ Model 1: unadjusted model. ^b.^ Model 2: adjusted for age, BMI, occupation, education level, household income, high-risk residential areas, smoking status, history of allergies, history of head trauma, family history of cancer, physical activity, and energy intake. ^c.^ Carotene per 500 μg/d increments, flavonoid per 10 mg/d increments, soy isoflavone and anthocyanin per 1 mg/d increments, resveratrol per 500 μg/d increments.

## Data Availability

The data presented in this study are available on request from the corresponding author.

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
