# Peer review of "Phytochemicals and Glioma: Results from Dietary Mixed Exposure"

_brainsci, 2023, doi:10.3390/brainsci13060902_

Round 1

Reviewer 1 Report

Phytochemicals and Glioma: Results from Dietary Mixed Exposure

Weichunbai Zhang et al.

In the manuscript "Phytochemicals and Glioma: Results from Dietary Mixed Exposure”  by Weichunbai Zhang et al.   the role of some phytochemicals in glioma development was analyzed. The authors claim that the higher intake of carotene, flavonoids, soy isoflavones, anthocyanin, and resveratrol is associated with a reduced risk of glioma. This statement may be true or not, as the presented data is not convincing. The manuscript misses some valuable information.

1)     There is no information how the amounts of   carotene, flavonoids, soy isoflavones, anthocyanin, and resveratrol were calculated (Figure 1)

2)     There is no data about the weights of the participants, only BMIs. The similar BMI does not mean the similar body weight. People with a bigger weight consume more food. More food means more phytochemicals. If the people in the control group have bigger weights, they could consume more phytochemicals than the case group. But the relative amounts (per kg of body weight) could be similar. So, it should be calculated the relative amounts (mg/kg) of phytochemical intakes.

English should be corrected

Author Response

Dear Reviewer:

The authors would like to express their gratitude for the positive and constructive comments and suggestions. We have revised the manuscript and would like to resubmit it for your consideration. The modifications were marked are marked in red font in this manuscript. Here are the point-by-point responses:

1)     There is no information how the amounts of carotene, flavonoids, soy isoflavones, anthocyanin, and resveratrol were calculated (Figure 1)

=> Responses: Thank you for your suggestion. We added information on calculating phytochemical intake. Please refer to Line 111-118 in the manuscript.

“The intake of phytochemicals was mainly calculated according to the dietary intake of individuals and the China food composition table. According to the intake frequency and intake of each food reported by the study subjects, the consumption of all foods was converted into daily intake (g or ml). The China Food Composition Table provided the unit amounts of carotene, flavonoids, soy isoflavones, anthocyanin, and resveratrol in each food. The total intake of this phytochemical is calculated by multiplying the daily intake of each food by the unit amount of each phytochemical in the food, and then summing the intake of each phytochemical in the different foods.”

2)     There is no data about the weights of the participants, only BMIs. The similar BMI does not mean the similar body weight. People with a bigger weight consume more food. More food means more phytochemicals. If the people in the control group have bigger weights, they could consume more phytochemicals than the case group. But the relative amounts (per kg of body weight) could be similar. So, it should be calculated the relative amounts (mg/kg) of phytochemical intakes.

=> Responses: Thank you for your suggestion. We calculated the relative intake of phytochemicals according to the body weight of the subjects, and the results are shown in Table S3, which is basically consistent with the original results. We supplement this result in the manuscript. Please refer to Line 144-146 and Line 256-258 in the manuscript and Table S3 in the supplementary materials.

“Sensitivity analysis was performed by calculating the phytochemical intake per unit body weight, and repeated logistic regression to avoid the influence of body weight difference.”

“The results of sensitivity analysis showed that the significance of the results was still consistent with the original results after replacing the original intake with the phytochemical intake per unit body weight.”

Reviewer 2 Report

The article presented by Zhang W. et al. entitled Phytochemicals and Glioma: Results from Dietary Mixed Exposure aimed to describe an epidemiological study related to different phytochemical intakes and the risk to develope gliomas in patients. In order to accomplish this, a case-control study was carry out. The article is interesting and well organized. I only have minor observations to adress.

1. The period of time were the study was carry out was not indicated. Please revise and add it.

2. Regarding study population, can the authors describe in more detail the characterstic of control group. Beside, explain the following sentence: Based on age (5 years) and gender, there was a 1:1 individual match.

3. What can the authors discuss in relation to "However, patients with gliomas had higher BMIs, lower educational levels, greater levels of smoking and physical activity, fewer allergies, and higher rates of cancer in their families than non-glioma patients." Could there be a bias in the choice of participants in the control group?

4. In mi opinion, discussion present a lot of information of in vitro or experimental assays that do not contribute to the present work. I suggest to summarize or delete it.

English quality is ok. A coherent readingof the article is allowed.

Author Response

Dear Reviewer

The authors would like to express their gratitude for the positive and constructive comments and suggestions. We have revised the manuscript and would like to resubmit it for your consideration. The modifications were marked are marked in red font in this manuscript. Here are the point-by-point responses:

  1. The period of time were the study was carry out was not indicated. Please revise and add it.

=> Responses: Thank you for your suggestion. We added the period of time in the manuscript. Please refer to Line 81 in the manuscript.

“We carried out this case-control study at Beijing Tiantan Hospital during 2022.“

  1. Regarding study population, can the authors describe in more detail the characterstic of control group. Beside, explain the following sentence: Based on age (5 years) and gender, there was a 1:1 individual match.

=> Responses: Thank you for your suggestion. We have supplemented the description of the control group and reinterpreted the sentence above. Please refer to Line 92-95 in the manuscript.

“The control group was healthy residents from the community who were also screened according to the inclusion and exclusion criteria described above and matched 1:1 with the case population by age (±5 years) and the same sex. “

  1. What can the authors discuss in relation to "However, patients with gliomas had higher BMIs, lower educational levels, greater levels of smoking and physical activity, fewer allergies, and higher rates of cancer in their families than non-glioma patients." Could there be a bias in the choice of participants in the control group?

=> Responses: Thank you for your suggestion. We have supplemented our discussion of this in the appropriate section. Please refer to Line 408-415 in the manuscript.

“Thirdly, there were significant differences in BMI, education, smoking, allergies, and family history of cancer between the case and control groups. It was considered to be related to other potential influencing factors of glioma. Existing studies have shown that glioma patients and healthy people were essentially different in smoking [68], BMI [76], allergies [77], family inheritance [78], and so on. This was similar to other study populations on diet and glioma [68]. To avoid potential confounding factors, we also adjusted these results in a multivariate model. Detailed subgroup analyses were per-formed for these variables, and the results remained relatively robust.”

  1. Mousavi, S.M.; Shayanfar, M.; Rigi, S.; Mohammad-Shirazi, M.; Sharifi, G.; Esmaillzadeh, A. Adherence to the Mediterranean dietary pattern in relation to glioma: A case-control study. Clin Nutr 2021, 40, 313-319, doi:10.1016/j.clnu.2020.05.022.
  2. Zhang, D.; Chen, J.; Wang, J.; Gong, S.; Jin, H.; Sheng, P.; Qi, X.; Lv, L.; Dong, Y.; Hou, L. Body mass index and risk of brain tumors: a systematic review and dose-response meta-analysis. Eur J Clin Nutr 2016, 70, 757-765, doi:10.1038/ejcn.2016.4.
  3. Turner, M.C.; Krewski, D.; Armstrong, B.K.; Chetrit, A.; Giles, G.G.; Hours, M.; McBride, M.L.; Parent, M.E.; Sadetzki, S.; Siemiatycki, J., et al. Allergy and brain tumors in the INTERPHONE study: pooled results from Australia, Canada, France, Israel, and New Zealand. Cancer Cause Control 2013, 24, 949-960, doi:10.1007/s10552-013-0171-7.
  4. Ostrom, Q.T.; Francis, S.S.; Barnholtz-Sloan, J.S. Epidemiology of Brain and Other CNS Tumors. Curr Neurol Neurosci 2021, 21, 68, doi:10.1007/s11910-021-01152-9.

  1. In mi opinion, discussion present a lot of information of in vitro or experimental assays that do not contribute to the present work. I suggest to summarize or delete it.

=> Responses: Thank you for your suggestion. We re-summarize the relevant content in the discussion section. Please refer to Line 389-400 in the manuscript.

“At present, the mechanism of phytochemicals in glioma is not clear. On the one hand, they participate in the regulation of apoptosis and autophagy, etc. Coelho et al. also found that flavonoid apigenin can decrease the survival rate and proliferation rate of rat C6 glioma cells in a time-dependent and dose-dependent manner by induc-ing differentiation, apoptosis, and autophagy [70]. Other phytochemicals, such as an-thocyanins [71] and resveratrol [72], had similar mechanisms. On the other hand, they played an antioxidant protective role in scavenging DNA-damaged free radicals and regulating DNA repair mechanisms [73,74]. In rats given 50 mg/kg naringenin orally for 30 days, it was discovered that lipid peroxidation was decreased, antioxidant ca-pacity was increased, and the expression of nuclear factor κB, etc was decreased in rat brain tissue. These changes significantly slowed the proliferation of glioma cells [75]. Therefore, they may still have potential preventive and therapeutic effects on gliomas.”

  1. Coelho, P.L.; Oliveira, M.N.; Da, S.A.; Pitanga, B.P.; Silva, V.D.; Faria, G.P.; Sampaio, G.P.; Costa, M.F.; Braga-de-Souza, S.; Costa, S.L. The flavonoid apigenin from Croton betulaster Mull inhibits proliferation, induces differentiation and regulates the inflammatory profile of glioma cells. Anti-Cancer Drug 2016, 27, 960-969, doi:10.1097/CAD.0000000000000413.
  2. Wang, L.S.; Stoner, G.D. Anthocyanins and their role in cancer prevention. Cancer Lett 2008, 269, 281-290, doi:10.1016/j.canlet.2008.05.020.
  3. Dadgostar, E.; Fallah, M.; Izadfar, F.; Heidari-Soureshjani, R.; Aschner, M.; Tamtaji, O.R.; Mirzaei, H. Therapeutic Potential of Resveratrol in the Treatment of Glioma: Insights into its Regulatory Mechanisms. Mini-Rev Med Chem 2021, 21, 2835-2847, doi:10.2174/1389557521666210406164758.
  4. Chen, Q.H.; Wu, B.K.; Pan, D.; Sang, L.X.; Chang, B. Beta-carotene and its protective effect on gastric cancer. World J Clin Cases 2021, 9, 6591-6607, doi:10.12998/wjcc.v9.i23.6591.
  5. Lee, J.Y.; Jo, Y.U.; Shin, H.; Lee, J.; Chae, S.U.; Bae, S.K.; Na, K. Anthocyanin-fucoidan nanocomplex for preventing carcinogen induced cancer: Enhanced absorption and stability. Int J Pharmaceut 2020, 586, 119597, doi:10.1016/j.ijpharm.2020.119597.
  6. Sabarinathan, D.; Mahalakshmi, P.; Vanisree, A.J. Naringenin, a flavanone inhibits the proliferation of cerebrally implanted C6 glioma cells in rats. Chem-Biol Interact 2011, 189, 26-36, doi:10.1016/j.cbi.2010.09.028.

Reviewer 3 Report

1. The introduction and discussion should be focused more on the observations and novelty of this study  according to related studies for example doi: 10.1186/s12937-021-00689-2

2. Exclusion criteria are also required 

3. More concluding remarks must be also added.

Minor language editing 

Author Response

Dear Reviewer

The authors would like to express their gratitude for the positive and constructive comments and suggestions. We have revised the manuscript and would like to resubmit it for your consideration. The modifications were marked are marked in red font in this manuscript. Here are the point-by-point responses:

  1. The introduction and discussion should be focused more on the observations and novelty of this study according to related studies for example doi: 10.1186/s12937-021-00689-2

=> Responses: Thank you for your suggestion.  We supplement it in the preface and discussion section. Please refer to Line 72-77 and Line 423-430 in the manuscript.

“Therefore, we quantified the association between five phytochemicals and glioma based on the dose-response relationship. Considering that dietary phytochemicals were not ingested as a single species [21], the association of dietary phytochemicals with glioma was assessed based on a co-exposure model to provide some evidence for phytochemicals to prevent gliomas to quantitatively evaluate the effect of dietary phytochemicals on glioma in the Chinese population.”

“Importantly, this was the first time that the association with glioma has been assessed in the context of mixed phytochemical intake. As the main source of phytochemicals, people did not take a single phytochemical, but a variety of phytochemicals were taken together through the dietary route. Therefore, the effect of co-exposure of these phytochemicals on gliomas had to be considered. Based on WQS regression and the BKMR model, we found the association between phytochemical co-exposure and glioma and further found that anthocyanins and carotene had the greatest effect on glioma among many phytochemicals.”

  1. Rigi, S.; Shayanfar, M.; Mousavi, S.M.; etc. Dietary phytochemical index in relation to risk of glioma: a case-control study in Iranian adults. Nutr J 2021, 20, 31, doi:10.1186/s12937-021-00689-2.

  1. Exclusion criteria are also required

=> Responses: Thank you for your suggestion. We added the corresponding exclusion criteria. Please refer to Line 89-92 in the manuscript.

“Exclusion criteria mainly included major eating behavior changes (such as weight loss), hormone use and other treatments that had an impact on nutrition, endocrine, diges-tive, and neurological diseases, abnormal energy expenditure (>5000 or 400 kcal/d), and pregnancy.”

  1. More concluding remarks must be also added.

=> Responses: Thank you for your suggestion. We rewrote the conclusion. Please refer to Line 432-439 in the manuscript.

“In conclusion, in this study of phytochemicals and glioma, we found that higher intakes of carotene, flavonoids, soy isoflavones, anthocyanins, and resveratrol were associated with a reduced risk of glioma, all of which had a significant dose-response relationship with glioma, and that the risk leveled off when intakes exceeded a certain amount. When phytochemical mix exposure was considered, a significant association with glioma was still demonstrated and was primarily caused by anthocyanins and carotene. Therefore, the effect of phytochemicals on glioma may be too significant to ignore. Future prospective studies should further confirm this relationship.”

Round 2

Reviewer 1 Report

No comments

Reviewer 3 Report

Accept

Accept